# Unusual Presentation of Bouveret Syndrome Resulting in Both Gastric Outlet Obstruction and Small Bowel Obstruction with Perforation

**DOI:** 10.3390/medicines9030024

**Published:** 2022-03-15

**Authors:** Jarod Shelton, Muhammad Adeel Samad, James Juhng, Shawn M. Terry

**Affiliations:** WellSpan Health, Department of Surgery, York, PA 17403, USA; msamad@wellspan.org (M.A.S.); jjuhng@wellspan.org (J.J.); sterry@wellspan.org (S.M.T.)

**Keywords:** Bouveret syndrome, gastric gallstone ileus, gallstone ileus, gallstone, choledochoduodenal fistula

## Abstract

Our case describes an 83-year-old female who presented with severe abdominal pain, nausea, and bilious emesis of one day’s duration. She had an endoscopic retrograde cholangiopancreatography (ERCP) with sphincterotomy and percutaneous transhepatic biliary drainage (PTCD) one year prior for choledocholithiasis with acute cholangitis in her home country, Scotland. Unfortunately, while visiting family in the United States, her PTCD became dislodged, and she developed progressive worsening abdominal pain. Computerized tomography of her abdomen showed pneumobilia, perigastric inflammation, a contracted gallbladder, small bowl inflammation with a likely transition point at the mid-jejunum, and a probable duodenal mass. The patient underwent an exploratory laparotomy with intraoperative findings of choledochoduodenal fistula with coincident gastric and small bowel obstruction (SBO) secondary to three large, mixed gallstones. One 3 cm gallstone was located at the pylorus and two (2.3 and 3 cm) gallstones were isolated in the mid-jejunum, with one of those causing isolated transmural pressure necrosis with subsequent perforation. Bouveret syndrome is a rare cause of gastric outlet obstruction (GOO) that manifests via an acquired cholecystoenteric fistula. Our patient presented with a concomitant GOO and SBO with perforation of the mid-jejunum. Timely diagnosis of Bouveret syndrome is essential, as most causes require emergent surgical intervention.

## 1. Introduction

Bouveret syndrome is a rare variant of gallstone ileus secondary to an acquired cholecystoenteric fistula that was initially described in 1770 by Beaussier; however, the eponym was not derived until Léon Bouveret extensively documented the disease in 1896 [1,2]. Bouveret syndrome is caused by a stone impacting itself in the stomach or duodenum, leading to a gastric outlet obstruction (GOO). Although the conditions that result in the development of Bouveret syndrome are relatively common, i.e., cholecystitis and cholelithiasis, the number of documented cases of Bouveret syndrome are fewer than 320 [3]. The literature has outlined how this rare subtype can cause significant morbidity and mortality rates greater than 30% [4]. Gallstones that migrate distally and become impacted at the ileocecal junction tend to cause SBO, which is characterized as gallstone ileus. The development of either condition can quickly become a surgical emergency that can lead to rapid clinical deterioration, especially if the obstructions are complete as opposed to partial [5]. We present the first case of Bouveret syndrome resulting in concomitant GOO and SBO with perforation. Bouveret syndrome has a high morbidity and mortality due to misdiagnosis and, therefore, the syndrome must be under consideration for any patient presenting with abdominal pain.

## 2. Case Description

We report the case of an 83-year-old woman who presented to our emergency department (ED) with progressive worsening abdominal pain, nausea, and non-bloody, bilious emesis of one day’s duration that continued overnight in the ED. Her medical history includes coronary artery disease status following angioplasty and stenting, in addition to choledocholithiasis with acute cholangitis requiring endoscopic retrograde cholangiopancreatography (ERCP) with sphincterotomy and percutaneous transhepatic biliary drainage (PTCD) approximately one year prior in her home country, Scotland. Regrettably, the patient’s PTCD tube became dislodged, and she began to develop severe abdominal pain, prompting presentation to our facility. There was no recent history of fever, jaundice, acholic stools, or dark urine. A two-view abdominal radiograph was initially obtained showing a branching density over the liver relating to portal venous gas (Figure 1). Computerized tomography (CT) of her abdomen showed circumferential mid-jejunum bowel wall thickening with a mildly dilated bowel proximally, prominent mesenteric edema, mild intra- and extrahepatic biliary ductal dilatation with pneumobilia, perigastric inflammation, a contracted gallbladder, and probable obstructing mass in the first portion of the duodenum (Figure 2). She developed peritoneal symptoms and was taken urgently for an exploratory laparotomy.

Intraoperative findings were consistent with Bouveret syndrome, with concomitant gastric outlet obstruction (GOO) and small bowel obstruction (SBO) caused by three large gallstones. A 3 cm, ovoid gallstone was found to be impacted and immobile within the lumen of the duodenum, and was extracted via a longitudinal pyloromyotomy and subsequently closed transversely in Heineke–Mikulicz fashion (Figure 3). Two large (2.3 and 3 cm) gallstones were palpated at the transition point in the mid-jejunum, which was edematous and inflamed (Figure 3). A single gallstone was impacted, causing transmural pressure necrosis with perforation. A 25 cm segment of the mid-jejunum was resected and anastomosed in side-to-side functional end-to-end fashion. The gallbladder appeared to be contracted with evidence of a choledochoduodenal fistula. Post-operatively, the patient had an uneventful recovery. An upper gastrointestinal tract radiograph was negative for extraluminal leak, and she was discharged on post-operative day 9. She was instructed to follow-up with her surgeon on her return to Scotland.

## 3. Discussion

Fistulation between the gallbladder and bowel is a multicomponent process that includes gallbladder wall inflammation, cholecysto-biliary adhesions, mechanical pressure exerted by gallstones, and ensuing ischemic injury [2,6]. The resultant cholecystoenteric fistula allows for the passage of large gallstones that are typically eliminated without consequence in feces or emesis until their diameter exceeds 2 cm [7]. Areas of either pre-existing stenosis or post-surgical anatomy are prone to gallstone impaction. Gallstones that migrate proximally or are released via a cholecystogastric fistula tend to cause GOO, which is termed Bouveret syndrome. Gallstones that are released and migrate distally have the potential to cause an SBO, which is termed a gallstone ileus.

Risk factors for Bouveret syndrome include female gender (ranging from 1.4:1 to 1.9:1), >60 years of age (median age of 74 years), large gallstones (>2 cm), recurrent cholecystitis, and post-surgical gastrointestinal anatomy [8]. The most common presentation of Bouveret syndrome can be characterized by Rigler’s triad: ectopic gallstone, GOO, and radiographic pneumobilia [2]. Imaging confirmed that gastric outlet obstruction secondary to an impacted gallstone is complicated by 85% of gallstones being radiolucent [9]. Furthermore, laboratory studies are relatively non-specific, with the most consistent finding being intravascular depletion [2,10]. The rarity and vague presentation of Bouveret syndrome commonly results in a delay in diagnosis, and when combined with the advanced age at presentation and age-associated comorbidities, the mortality rate has been reported to be as high as 37% [11].

The development of a definitive diagnosis and treatment algorithm is complicated due to the variable presentation of Bouveret syndrome. Khan et al. [11] proposed a treatment algorithm once a diagnosis has been established. All presentations of GOO should be managed conservatively with nasogastric tube decompression, fluid resuscitation, and electrolyte correction. Lithotripsy by esophagogastroduodenoscopy can first be attempted in stable patients. If lithotripsy is unsuccessful or the patient decompensates, then enterolithotomy should be performed. The enterolithotomy considerations include a one-stage versus two-stage approach of lithotomy with cholecystectomy and fistula repair performed at the index operation versus later due to recurrent biliary complications, respectively. Consideration for cholecystectomy with fistula repair should be performed on a case-by-case basis depending on the patient’s clinical presentation.

## 4. Conclusions

We present the first case of Bouveret syndrome that resulted in concomitant GOO and SBO. Our patient was of the appropriate demographic and had a history of gallbladder disease. Her presentation to our facility was relatively non-specific; however, her prior history of choledocholithiasis with ERCP and PTCD was concerning for gallbladder pathology. Imaging demonstrated pneumobilia, perigastric inflammation, SBO, and probable obstructing mass in the first portion of the duodenum. During the initial work-up, the patient clinically deteriorated and was taken emergently for an exploratory laparotomy. Three large, mixed gallstones were identified to be causing both GOO and SBO with perforation. A choledochoduodenal fistula was identified, suggesting that the gallstones migrated both proximally and distally. In our case, the diagnosis of Bouveret syndrome was suspected pre-operatively, and timely intervention resulted in a positive patient outcome.

## 5. Lessons Learned

Bouveret syndrome is a rare cause of gastric outlet obstruction that manifests via an acquired cholecystoenteric fistula. The pre-operative diagnosis of Bouveret syndrome can be difficult in an unstable patient due to the non-specific nature of the disease. Simultaneous gallstone ileus is possible. Regardless, timely intervention is essential to ensure positive patient outcomes.

## Figures and Tables

**Figure 1 medicines-09-00024-f001:**
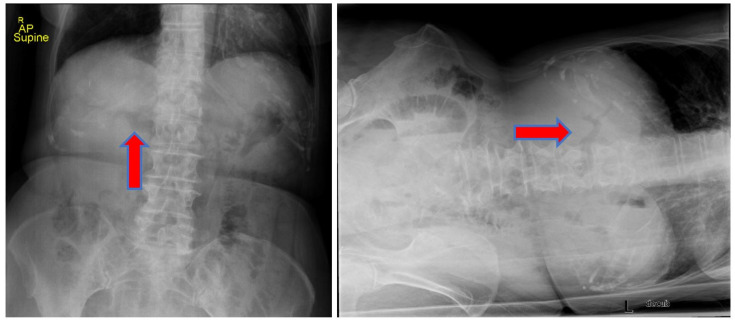
Abdominal radiograph showing branching density over the liver indicating portal venous gas (red arrow). The figure on the **left** is a supine anteroposterior (AP) chest view and on the **right** is a left lateral decubitus chest view.

**Figure 2 medicines-09-00024-f002:**
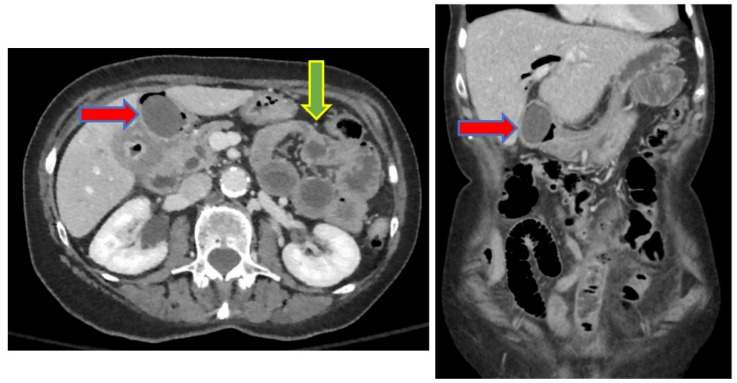
CT abdomen showing duodenal obstructing mass (red arrow) and mid-jejunum transition point (green arrow). The figure on the **left** is an axial slice and on the **right** is a coronal slice.

**Figure 3 medicines-09-00024-f003:**
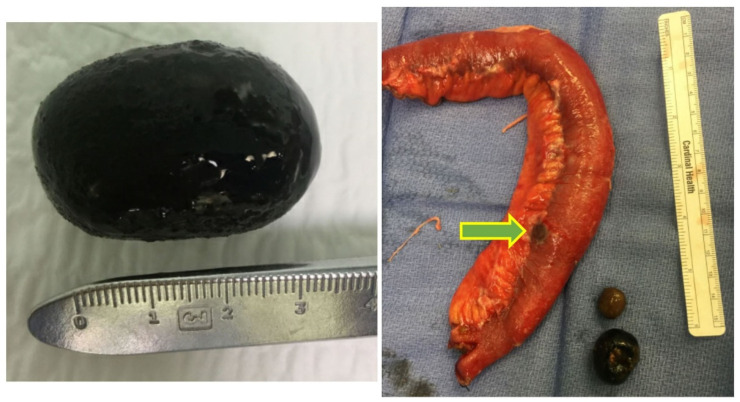
On the **left**, 3 cm gallstone extracted from pylorus. On the **right**, resected mid-jejunum with transmural pressure necrosis (green arrow) with extracted gallstones (2.3 and 3 cm).

## Data Availability

Not applicable.

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
