# Peer review of "Unusual Presentation of Bouveret Syndrome Resulting in Both Gastric Outlet Obstruction and Small Bowel Obstruction with Perforation"

_medicines, 2022, doi:10.3390/medicines9030024_

Round 1

Reviewer 1 Report

The present study aimed at presenting a case report of a patient presenting the first case of Bouveret syndrome that resulted in both a gastric outlet syndrome and small bowel obstruction. This is an overall well-written case report study reporting a rare disease whose diagnosis and treatment may be complicated to establish despite a major unclear point.

The authors acclaim to present the first double gastric/small bowel obstruction syndrome although this observation is not well documented. Indeed, neither in figure 1 nor in figure 2 do we see a gastric outlet syndrome present. The stomach does not present any distension and a nasogastric tube is absent. Thus, presence of a gallstone in the first portion of the duodenum seems more likely to be an incidental finding rather than a true Bouveret syndrome. Patient vomiting could be more likely related to small bowel obstruction than a gastric outlet syndrome. Please provide imaging findings supporting your case report. 

Reviewer 2 Report

The manuscript describes a first-time case report based on Bourveret syndrome resulting in both gastric outlet obstruction and small bowel obstruction with perforation. The authors examined important/uncommon clinical scenarios to illustrate and inform the use of clinical guidelines. Also, the authors succinctly summarized to provide a key clinical message. 

Minor points:

  1. Please start the abstract with one/two sentences about the significance of the medical condition(s) the case report is about before the existing abstract. By the way, summarizing the whole case in the abstract was not necessary.
  2. The introduction written in the manuscript is too short to have any general idea about the condition or symptoms related to the case report. It should be elaborated with the necessary reference. Also, it should answer the question that why do the authors think the case is important. If needed authors can shift some material from discussion to introduction.

Round 2

Reviewer 1 Report

accept in present form

Author Response

The requested changes have been made. Thank you.